# Multimodal Prehabilitation for Gynecologic Cancer Surgery

**DOI:** 10.3390/curroncol32020109

**Published:** 2025-02-14

**Authors:** Jeongyun Kim, Chae Hyeong Lee, Ga Won Yim

**Affiliations:** Department of Obstetrics and Gynecology, Dongguk University College of Medicine, Goyang 10326, Republic of Korea; jyun@dumc.or.kr (J.K.); gynecancer@dumc.or.kr (C.H.L.)

**Keywords:** cancer, frailty, gynecologic surgery, postoperative complications, prehabilitation, preoperative care

## Abstract

Surgical treatment is commonly employed to treat patients with gynecologic cancer, although surgery itself may function as a stressor, reducing the patients’ functional capacity and recovery. Prehabilitation programs attempt to improve patients’ overall health and baseline function prior to surgery, thereby enhancing recovery and lowering morbidity. In recent years, prehabilitation has come to primarily refer to multimodal programs that combine physical activity, nutritional support, psychological well-being, and other medical interventions. However, the specific methods of implementing prehabilitation and measuring its effectiveness are heterogeneous. Moreover, high-level evidence regarding prehabilitation in gynecologic cancer surgery is limited. This review provides a summary of multimodal prehabilitation studies in gynecologic oncologic surgery. Enhanced postoperative recovery, lower postoperative complications, lower rate of blood transfusions, and faster gastrointestinal functional recovery have been reported after multimodal prehabilitation interventions. Patients and healthcare professionals should recognize the importance of prehabilitation in the field of gynecologic oncologic treatment, based on the emerging evidence. In addition, there is a need to establish an appropriate target group and construct a well-designed and tailored prehabilitation program.

## 1. Introduction

Cancer is one of the leading causes of death worldwide, and causes economic and social burdens [1]. Among cases of cancer and associated deaths, gynecologic cancers account for a significant portion (7.3% of incidence and 7.0% of mortality globally) [1]. In 2022, there were 661,021 newly diagnosed cervical cancer cases, with 348,189 deaths. For uterine corpus cancer, 420,242 cases were newly identified, with 97,704 deaths, and 324,398 new cases of ovarian cancer were diagnosed, with 206,839 deaths [1]. A substantial proportion of these gynecologic cancer patients need surgical treatment, and surgery itself causes metabolic and physiological stress [2]. It has been reported that the complication rate after major surgery is more than 30%, with a decrease in physiological reserve by 40%. Also, the patient’s functional ability does not fully recover until 6 to 9 weeks after surgery [3]. Moreover, gynecologic cancers pose a significant problem in that a considerable proportion of gynecologic cancer patients are of advanced age, exhibit obesity, lead a sedentary lifestyle, experience malnourishment, and show depressive symptoms [4,5]. These factors may contribute to slow recovery after surgery and increase complications, necessitating that these modifiable factors are addressed prior to surgical treatment.

Prehabilitation programs aim to accelerate postoperative recovery and reduce complications by identifying and correcting problems before surgery, as well as improving the functional capacity and metabolic reserves of the patient. In the oncologic field, implementing targeted interventions to ‘prehabilitate’ the patient within the continuum of cancer treatment is a new, emerging field of care delivery [6]. Therefore, there have been recent attempts to offer prehabilitation programs in conjunction with, or as a part of, enhanced recovery after surgery (ERAS) protocols [7]. The current ERAS guidelines mainly deal with perioperative management during hospitalization for an operation, especially focusing on postoperative recovery, and not addressing presurgical risk factors [8,9]. Prehabilitation, on the other hand, aims to maximize the patient’s condition for surgery prior to hospitalization. More evidence has been elucidated regarding the integration of preoperative optimization through the prehabilitation program, along with surgical stress minimization through the ERAS program [7,10]. Recent studies focus on multimodal or trimodal prehabilitation programs that seek to improve physical, nutritional, and psychological aspects before surgery and expect the benefits of their interaction. Therefore, recent interventions mainly adopt multimodal strategies due to better functional outcomes compared to a single modality [7,11].

Despite the increasing number of feasibility studies and investigations, prehabilitation programs have no uniform recommendations regarding the composition and intensity of such interventions. Moreover, heterogeneity exists in various prehabilitation programs due to the lack of high-quality evidence, especially in the field of gynecologic oncology [12,13]. This review describes current research on multimodal prehabilitation interventions in patients with gynecologic cancer. We aim to summarize whether intervention in each area of prehabilitation is effective and leads to enhanced postoperative recovery and fewer complications.

## 2. Methods

For this narrative review, a literature search was performed utilizing the PubMed database. To include and discuss prehabilitation studies in the field of gynecologic oncology, we used the search terms ‘prehabilitation’ and ‘gynecologic cancer or ovarian cancer or endometrial cancer or cervical cancer’. Only English-language studies and studies that involved multimodal prehabilitation interventions were included for analysis. Although we conducted a comprehensive search of articles published from the database’s inception through November 2024, our primary focus was to identify studies that targeted the recent trend of trimodal prehabilitation, including exercise, nutritional support, and psychological intervention. Case reports, review articles, and studies that only dealt with ERAS guidelines, single intervention, or were unrelated to gynecologic cancers were excluded. However, studies that covered a broad range of surgical treatments for mixed cancers, including gynecologic cancer, were included in this study. The details of the study selection process are illustrated in Figure 1.

Observational studies and randomized interventional studies were reviewed with regard to the following aspects: procedures and compliance, improvements after intervention, length of hospital stay, postoperative recovery, and perioperative complications.

## 3. Results

Six studies on multimodal prehabilitation intervention related to gynecologic oncologic surgery were identified, and are summarized in Table 1. There was one randomized interventional study, two prospective cohort studies, two retrospective cohort studies, and one mixed cohort study. All studies were single-center studies that applied multimodal programs, including physical exercise, nutrition, psychological support, and other medical interventions. Five of these studies adopted the ERAS program, whereas the remaining study included patients with peritoneal carcinomatosis undergoing cytoreductive surgery (CRS) and hyperthermic intraperitoneal chemotherapy (HIPEC) without ERAS [14]. All studies were conducted in Spain, except one Polish [15] and one Chinese study [16]. The only prospective randomized study in the gynecologic oncologic field to date has been published by the Polish group, Zebalski et al. [15]. They compared 36 patients who received prehabilitation before surgery for confirmed or suspected ovarian cancer with 34 patients who did not. The prehabilitation program consisted of resistance and aerobic exercises (physical), diet and protein/immunomodulator supplementation (nutrition), and referral to psychologists upon screening (psychological), along with optimization of laboratory test results and counseling on risky behaviors prior to surgery. Although the primary outcome of this study was to investigate the effect of prehabilitation on the postoperative complication rates and hospitalization length after ovarian cytoreductive surgery, the main objective was to verify whether an electromyography device could be used as a marker of patient compliance by measuring muscle strength and tension in the abdominal rectus muscles during the program. According to their study, the prehabilitation intervention showed a positive effect on physical capacity, measured by a 6 min walking test (6MWT) and muscle tension. However, there was no difference in the impact on quality of life between the two groups.

## 4. Prehabilitation in Gynecologic Oncology

### 4.1. Physical Intervention

Adequate preoperative physical activity and inspiratory muscle strength are known to be associated with improved postoperative outcomes [19]. These have also been major elements of prehabilitation intervention. Aerobic, resistance, and respiratory exercises are commonly used, and some studies incorporate flexibility exercises [10], stretching, and Kegel exercises [16] as their physical intervention. Exercise interventions are mainly conducted as either home-based or supervised programs, but some studies have attempted hospital-based interventions to attain short-term effects within an average of 7 days [16]. Information booklets, videos, and mobile-based applications are also used when necessary. Exercise intensity is adjusted to low, medium, and high depending on the patient’s physical condition [14], while another study recommends a gradual increase in intensity [15]. Exercise frequency (daily to three times per week), intensity, and duration (two to four weeks) differ between studies.

The 6MWT is the most common tool used for evaluating patients’ physical capabilities. This functional status test can be conducted with ease and safety in an outpatient clinic setting [20]. Other methods used for evaluation include the Yale Physical Activity Survey [9] and maximal oxygen consumption (VO2 max) [10,15]. Using these assessment tools, physical preparation before gynecologic cancer surgery has shown enhanced physical capacity after intervention.

A recent prospective, single-center study by Lario-Perez et al. [14] reported improved functional walking capacity after aerobic and muscular resistance training and relaxation exercises in 62 patients with peritoneal cancer. An improvement of 42.2 m (interquartile range [IQR] 39.62–44.72) in the 6MWT was shown on a treadmill the day before surgery, compared to baseline data, after a median of 27 days (IQR 19–34) of home-based intervention. This program’s compliance ratio for aerobic exercise and resistance exercise was 0.89 and 0.74, respectively. The 6MWT was the only independent risk factor for 90-day postoperative morbidity in the study. Comparable results were confirmed in a Chinese prospective cohort, in which a hospital-based program including stretching, resistance, aerobic, respiratory, and Kegel exercises was implemented for gynecologic cancer patients during a mean of 7.2 days (standard deviation [SD] 1.6) [16]. The compliance rate was 100%, and a difference in physical status was reported not only on the day before surgery, but also on the 30th day after surgery. In the intervention group, the results of the 6MWT improved in 37 (77.5%) patients and increased by 38.27 m on the day before surgery, and thereafter decreased by 51.49 m on the 30th day after surgery, resulting in a decrease of 13.23 m from baseline. In addition, 35 (71.4%) recovered their walking capacity to the level recorded on the day of hospitalization. On the contrary, in the control group, the results of the 6MWT decreased by 15.15 m the day before surgery and by 66.5 m on the 30th day after surgery, showing a total decrease of 81.64 m compared to baseline. Among the individuals in this group, 16 (33.3%) had reduced walking capacity the day before surgery, and 47 (97.9%) on the 30th day after surgery. Similarly, the prospective randomized interventional study by Zebalski et al. [15] conducted exercise intervention for an average of 24.5 days (SD 26.9) for patients with ovarian cancer, and also showed a median increase of 17 m (IQR 0–42.5, *p* < 0.001) in the results of the 6MWT compared to the control group on the day of admission. However, the maximum rate of oxygen consumption (VO2 max) during physical exertion, which is a measurement of physical fitness, showed no difference between the intervention and control groups. These findings underscore the need for further research regarding the optimal duration and intensity of physical intervention before surgery.

### 4.2. Nutritional Intervention

Weight loss is prevalent among cancer patients, and given that cancer impairs physical activity by causing malnutrition and sarcopenia, appropriate nutritional support is crucial [7]. In addition, while nutritional support enhances the efficacy of resistance training and accelerates an increase in muscle tone, without adequate protein intake, functional capacity may decrease due to muscle wasting [21,22]. Nutritional intervention has also been reported to reduce the length of hospital stay, intestinal recovery time, and perioperative complications [23]. Nutritional intervention programs prior to surgery commonly include daily protein intake (1.2–2.0 g/kg of body weight), supplementation with whey protein, and an immunomodulatory formula. Oral protein supplements are recommended to be taken 30 min after exercise to enhance muscle hypertrophy [17]. Patients are instructed to limit carbohydrates, particularly simple carbohydrates, eliminate processed foods, and increase vegetables rich in vitamins [15]. Some hospital-based programs provide a diet with a carbohydrate:protein:fat ratio of 6:3:1 [16]. In all programs, a dietitian or nutritionist is involved to assess patients and optimize the dietary plan.

Methods used to evaluate patients’ status include the Global Leadership Initiative on Malnutrition criteria, the MNA (mini nutritional assessment) and MUST (malnutrition universal screening tool) questionnaires, nutritional risk screening—2002, bioimpedance analysis (appendicular skeletal muscle index [ASMI]), body mass index, skeletal muscle mass using computed tomography (CT), and blood tests (total protein, albumin, hemoglobin, and prealbumin levels).

The usage of blood test results to assess patient compliance and the duration of nutritional intervention is inconclusive. In a prehabilitation study that provided dietitian-led nutritional optimization by various supplementations in patients with ovarian cancer, preoperative prealbumin levels were higher in patients with 100% compliance than in the control cohort after a median of 2 weeks (IQR 2–3) [9]. Another study found higher mean preoperative and postoperative total protein levels and higher postoperative albumin levels compared to controls after a mean of 13.5 weeks (SD 2.0) of nutritional counseling intervention without assessing compliance [17]. In this study by Miralpeix et al., patients with ovarian cancer were given homemade recipes for protein supplementation during the course of neoadjuvant chemotherapy before interval debulking surgery. In contrast, there were no significant differences in total protein and albumin concentrations among prehabilitation group patients after an average of 24.5 days (SD 26.9) of diet and protein supplements before ovarian cancer debulking surgery in the randomized study by Zebalski et al. [15]. The authors assumed that the relatively short prehabilitation period would not result in noticeable changes in laboratory parameters. Further research is needed to determine whether total protein levels can be used as a marker for nutritional improvement during prehabilitation programs.

Compliance and patient adherence are other critical issues that must be considered when implementing a prehabilitation program, especially when incorporating nutrition and exercise interventions. In the prospective cohort study of patients with peritoneal carcinomatosis, 95% compliance with nutritional supplement intake and lower compliance with physical exercise (89% with aerobic exercise and 74% with resistance exercise) yielded no improvement in the ASMI level after a median of 24 days of intervention [14]. Likewise, bioelectrical impedance analysis, as well as serum laboratory marker changes, can be used to assess the effectiveness of intervention and patient adherence. Interestingly, all nutritional interventions to date have included only supplementation or diet coaching. However, for gynecologic cancer patients for whom enteral nutrition is unsuitable at the time of initiating cancer treatment, total parenteral nutrition (TPN) may be beneficial as a means of prehabilitation as well [23]. This is particularly the case in advanced ovarian cancer patients, since they have a high prevalence of malnutrition. However, the current evidence regarding the use of TPN as a process of prehabilitation is lacking.

Despite the potential benefits of prehabilitation interventions, outcomes will vary according to patient compliance and acceptance. The most common reasons for declining participation are known to be related to program intensity and lack of motivation [7]. Therefore, flexible and tailored approaches, along with objective assessment tools, are needed. In addition, there is a need to screen and assess vulnerable patients who possess greater demands for improvement, rather than routinely implementing prehabilitation interventions for all patients, especially if resources are limited.

### 4.3. Psychological Intervention

Patients frequently encounter psychological problems such as depression and anxiety after a cancer diagnosis, affecting the course of cancer treatment [24]. Psychological factors are known to influence postoperative recovery and pain by impacting the pathophysiological mechanisms associated with the surgical stress response [25]. This is an important aspect of prehabilitation programs, which evaluate outcomes based on postoperative length of stay and complications. Also, the psychological intervention itself may motivate patients to actively engage in other elements of prehabilitation, such as physical and nutritional interventions [10].

All six studies identified in this review commonly adopted the hospital anxiety and depression scale (HADS) as a tool to evaluate patients’ psychological status. Patients were referred to a specialist, if needed, after the initial screening and evaluation. Other psychological interventions included relaxation and breathing exercises [10,14], yoga relaxation techniques, and meditation [16].

According to the studies conducted to date, the outcomes of psychological interventions are controversial. Following a median of 17 days (IQR 11–26) of relaxation exercises within a median of 27 days (IQR 19–34) of prehabilitation, no significant differences were observed in the HADS scores between baseline and one day before CRS for patients with peritoneal carcinomatosis [14]. Similarly, patients with ovarian cancer who were scheduled for CRS were referred to a psychologist based on a HADS score of 8 or provided counseling upon request, and this resulted in no significant differences compared to the initial value of the HADS score [15]. An average of 24.5 days of prehabilitation program was given, without any indication of compliance. In this study, quality of life was assessed using the standardized EORTC Quality of Life QLQ-C30 questionnaire, which also did not show significant improvement. Conversely, a hospital-based program for gynecologic cancer patients that consisted of coping strategies to reduce anxiety (yoga relaxation and psychologic consultation) showed a 100% compliance rate for a mean of 7.2 days, and improvement in the hospital anxiety scale on the day before surgery and the 30th day after surgery [16]. They also assessed the overall health level evaluated by the RAND 12-Item Health Survey v2 (SF-12v2), which increased from baseline to the day before surgery, but decreased on the 30th day after surgery, in the intervention group. There was no difference in baseline values compared to the control group. However, the values of the control group continued to decrease compared to the intervention group, demonstrating that the intervention helped overall health recovery. Thus, these data indicate that psychological interventions could help to alleviate anxiety compared to the control group.

Additionally, there were two studies that conducted group sessions for psychological intervention: cognitive behavioral group sessions [9] and supervised group mindfulness hospital sessions [10]. One of them further replaced group sessions with individual online sessions during the COVID-19 outbreak, and demonstrated an 80% compliance rate with the psychological intervention [9]. Care delivery using mobile or online platforms may be feasible, although there is no robust evidence of the effectiveness outcomes for these types of prehabilitation.

Psychological intervention is assumed to be an factor that apparently helps to improve postoperative outcomes and the effectiveness of prehabilitation. However, the impact will be affected by the patient’s underlying condition, the course of the treatment, and the disease itself, complicating the use of standardized intervention approaches and assessments. Furthermore, studies focusing on gynecologic oncology patients, let alone gynecologic patients [26], are still too lacking to establish appropriate psychological interventional programs.

### 4.4. Frailty Assessment and Medical Optimization

Frailty is a clinical condition defined by excessive vulnerability of an individual to both endogenous and exogenous stressors [27]. The objective of prehabilitation is to evaluate frailty prior to surgery and improve reserves by actively implementing interventions for the vulnerable group. Diverse methods to evaluate frailty have been developed and are currently utilized for patients with gynecologic oncology: the Charlson Comorbidity Index [9], the Eastern Cooperative Oncology Group performance status [9], the Geriatric 8 score [10,15], and the American Society of Anesthesiologists classification [18]. Studies conducted to date on prehabilitation interventions for gynecologic oncology patients have been focused on comparing frailty between an intervention group and a control group, or referring patients to a geriatrician when deemed essential [10]. However, to prioritize individuals requiring prehabilitation with limited resources, it is necessary to confirm the necessity and the role of intervention according to an individual’s frailty level. Subsequent studies need to include a comprehensive analysis of this vulnerable group.

Two studies have separately listed medical optimization in their prehabilitation programs, including correction of abnormal laboratory results, cessation of smoking and drinking, and management of comorbidities (hypertension, chronic obstructive pulmonary disease, chronic heart disease, and diabetes) [10,15]. One of them indicated that patients in the prehabilitation group needed significantly fewer packed red blood cell transfusions after surgery than the control group, despite the fact that oral iron supplementation had no impact on hemoglobin levels [15]. Moreover, a decreased hemoglobin concentration correlated with an extended postoperative hospital stay (*p* = 0.045).

The ERAS protocol for gynecologic oncology recommends the cessation of alcohol and smoking, as well as the correction of anemia, as components of prehabilitation [28]. Nonetheless, there is insufficient evidence concerning the success rate of these interventions. Moreover, data are scarce regarding which approach would be the most optimal and effective method that would ultimately contribute to enhancing postoperative recovery. Further research is necessary to design an ideal multimodal prehabilitation program for gynecologic oncologic patients.

### 4.5. Postoperative Outcomes

The expected outcomes of prehabilitation are faster recovery, as well as decreased length of hospital stay and postoperative complications. The findings on the length of postoperative hospital stay vary among studies: three studies reported a shortened length of stay [9,15,18], and the other three described no significant change [14,16,17]. In regard to postoperative complications, most of the studies used the Clavien–Dindo classification and the comprehensive complication index (CCI).

In a prospective randomized study by Zebalski et al. [15], patients who underwent CRS for ovarian cancer showed a shorter hospital stay of 2 days (median 5.0 days, IQR 4.0–6.2 vs. 7.0 days, IQR 6.0–10.0 in the control group; *p* < 0.001). A negative correlation was seen between maximum muscle tension and the duration of postoperative hospitalization (R = −0.35, *p* = 0.039). Also, the prehabilitation group had fewer complications according to the Clavien–Dindo classification (without complications, 47.2% vs. 20.6% in the control group; *p* = 0.02) and fewer postoperative transfusions (14% vs. 47% in the control group; *p* = 0.002). Another study by Diaz-Feijoo et al. [9] reported shortened postoperative hospitalization by a median of 2 days (5 days, IQR 4–6 vs. 7 days, IQR 5–9 in the control group; *p* = 0.04) in patients with advanced ovarian cancer after trimodal prehabilitation. Interestingly, the time to the start of chemotherapy was also shortened by a median of 10 days (25 days, IQR 23–25 vs. 35 days, IQR 28–45 in the control group; *p* = 0.03). Although the mean CCI scores were not significantly different between the prehabilitation group and their historical control group, Clavien–Dindo grade III complications were more common in the control group (0% vs. 26.3% in the control group). A retrospective cohort study by Miralpeix et al. [18], analyzing patients with endometrial cancer who underwent laparoscopic surgery and received the perioperative care of ERAS and a prehabilitation program, demonstrated a reduced hospital stay of 1 day (median 2.0 days, range 1–4 vs. 3.0 days, range 1–10 in the control group; *p* < 0.001) and an earlier normal oral diet start time of 3.6 h (mean 10.2 h, SD 4.3 vs. 13.8 h, SD 5.8 in the control group; *p* = 0.005). The objective of the study was to elucidate whether prehabilitation enhances surgical outcomes when combined with the usual ERAS program compared to ERAS alone. Despite the other perioperative improvements mentioned above, patients in both groups had similar postoperative complications (Clavien–Dindo grade > II, 7.4% vs. 5% in the control group; *p* = 0.58) and readmission rates (2.9% vs. 1.7% in the control group; *p* = 0.63). It is unclear why perioperative outcomes do not translate to decreased complication rates in some studies. In a prehabilitation study for patients with ovarian cancer receiving neoadjuvant chemotherapy and scheduled for interval CRS, there were no significant differences in postoperative complications (according to the Clavien–Dindo classification), pain control, and readmission rates between the groups [17]. However, the prehabilitation group demonstrated better intraoperative complications (14.3% vs. 40% in the control group) and lower intraoperative blood transfusion rates (14.3% vs. 53.3% in the control group), as well as a trend toward lower intraoperative vasoactive drug requirement (7.1% vs. 20.0% in the control group), suggesting improved tolerability during surgery. Differences in these outcomes may be due to compliance and variations in the baseline physical status of patients, which are known to be significantly influenced by the program’s design [17,29].

In a prospective study of patients with peritoneal carcinomatosis who received home-based prehabilitation and underwent CRS, surgical outcomes were compared based on the results of the 6MWT on a treadmill the day before surgery [14]. Although there was no significant difference in length of stay between the two groups, patients who could walk more than 360 m in the 6MWT on a treadmill after prehabilitation were less likely to have Clavien–Dindo II–V postoperative complications (*p* = 0.016). Notably, walking less than 360 m in the 6MWT on a treadmill after prehabilitation was considered the only independent risk factor for Clavien–Dindo II–V postoperative morbidity (odds ratio 3.99, 95% confidence interval 1.01–15.79; *p* = 0.048).

Several studies have confirmed the effectiveness of prehabilitation intervention on postoperative outcomes, although the evidence is still lacking. There may be limitations not only from the heterogeneous programs of each study, but also from the influence of the ERAS program that has already been applied. Moreover, the radicality of surgery in the case of ovarian cancer may alter postoperative complication rates and severity.

### 4.6. Challenges and Limitations of Prehabilitation in Gynecologic Oncology

Likewise, prehabilitation studies in the field of gynecologic oncology showed considerable heterogeneity in intervention methods and outcome measures. Nevertheless, the majority of studies reported favorable results, and further beneficial findings are expected in the years to come when considering the encouraging outcomes of ERAS. All six studies identified for this review confirmed improvements in intraoperative blood transfusion, postoperative recovery and complications, and initiation time of normal diet, first ambulation, gastrointestinal recovery, and chemotherapy, respectively. Other differences in outcomes may stem from varied intervention settings and small sample sizes.

Small observational studies addressing the feasibility and acceptability of intervention showed that prehabilitation in gynecologic oncologic patients was received positively by patients, and should be actively considered [30,31]. Furthermore, the type of gynecologic cancer may influence the plans and outcomes of prehabilitation programs compared to other types of cancer. For example, a significant number of patients with endometrial cancer are obese, which is a factor that requires attention in regard to intraoperative and postoperative complications [32,33,34]. Also, previous studies have shown that women have a higher rate of sarcopenia [14] and that gynecologic oncology patients tend to have a sedentary lifestyle [30]. In the case of ovarian cancer, there is a higher chance of patients being of old age, undergoing ultraradical surgery, and undergoing neoadjuvant chemotherapy, which necessitates active enrollment in the prehabilitation program [35]. In addition, financial toxicity for both the patient and society is of paramount importance when discussing cancer treatment. Dholakia et al. [36] reported the cost-effectiveness of prehabilitation relative to usual care alone for medically frail patients undergoing primary CRS for ovarian cancer in 2021. They showed a favorable result of prehabilitation by calculating cost-effectiveness of up to $9418 per patient after reducing severe complications, mortality, and non-home discharges. It is obvious that the total cost of treatment will increase in proportion with increased toxicity and/or complications. Therefore, many aspects of prehabilitation, including cost-effectiveness analysis, should be studied in order to apply this concept to the continuum of cancer care. Additionally, Dholakia et al. assumed the cost-effectiveness of their hypothetical cohort, predicting a reduction in non-home discharges following prehabilitation. However, empirical data on this topic are currently lacking, and further trials are necessary to evaluate the actual impact of non-home discharge rates on clinical outcomes.

Despite growing insights into the concept of prehabilitation, it has not been actively implemented in real practice. In a questionnaire-based survey of 206 healthcare professionals and 19 hospital trusts, 59% considered frailty scores in treatment decisions, and 83% received additional information about performance scores through frailty scores [37]. However, 63% of institutions did not routinely screen for frailty, and only 16% had a dedicated pathway for preoptimization of patients with frailty. Hence, it is crucial to develop a consensus on guidelines and evaluation methods in order to facilitate effective interventions, as demonstrated by the ERAS guidelines. This requires establishment by forthcoming randomized trials. Furthermore, prehabilitation issues should be addressed across all cancer types, including rare cancers such as vaginal and vulva cancer. To date, there is only one study that has investigated the potential benefits of a daily hip exercise regimen during pelvic radiation therapy for patients, including vaginal cancer [38]. However, to our knowledge, no study has yet focused on the effects of prehabilitation in patients undergoing surgical treatment for vulvar or vaginal cancer. This gap in the literature may be attributed to the small sample size of these patient populations, indicating the need for further research.

Table 2 summarizes the currently ongoing randomized controlled trials [39,40,41,42]. The largest ongoing randomized clinical trial is by Lopes and colleagues from Brazil (PROPER trial) [39]. They hypothesize that a multidisciplinary, supervised trimodal prehabilitation program in patients who are to undergo open gynecologic cancer surgery will lead to a reduction in the length of hospital stay and improve the patients’ functional capacity. A total of 194 patients will be randomized 1:1 to either prehabilitation plus an ERAS protocol or ERAS alone. Participants will receive at least 2–3 weeks of prehabilitation interventions, which will consist of physiotherapy (6–9 sessions of aerobic, inspiratory, and muscle strengthening) and nutritional counseling, in order to design a realistic prehabilitation program, especially for those who already have existing ERAS protocols.

Challenges facing the implementation of prehabilitation into current practice include cost, educating patients, the requirement of a multidisciplinary team, and the need for simplicity in terms of the intervention itself. Several strategies have been proposed to increase efficiency and reduce costs for interventions, such as online training, the use of wearable devices, phone consultation, collaboration with nearby local facilities, and feedback or incentives. Face-to-face interventions may be more effective, but other appropriate alternatives may help to increase patient compliance when patient preferences and costs are taken into consideration [31,43]. Timely referral to local organizations may be a strategy when the intervention of professional personnel is necessary and when patient motivation is demanded [44]. The low motivation of patients due to time constraints and a lack of social support are also a few of the common obstacles to implementing preoperative prehabilitation in cancer patients. Furthermore, patients may not want to postpone surgery by additional interventions, with concerns about their oncologic outcomes [44]. However, Li et al. [16] demonstrated that short-term interventions of seven days can also provide favorable outcomes, and suggested the need to establish efficient, short-term programs. Also, the juggle between the duration/intensity and effectiveness of prehabilitation programs should consider patient compliance [7]. It has been reported that higher program intensity may hinder compliance, particularly among elderly patients who are unaccustomed to physical activity [7]. Interestingly, however, patient motivation may be heightened by group activities, regardless of program intensity. In a study of physical exercise adherence among elderly patients, the retention rate of patients was 90% in the peer group, therefore suggesting the patients’ desire to connect and share mutual support with other patients facing similar challenges [45,46].

## 5. Conclusions

Preoperative physical, nutritional, and psychological preparedness before cancer surgery is of paramount importance, since emerging data suggest that appropriate interventions can help patients to cope better with surgical stress by improving their physiological and psychological reserves. Although prehabilitation has been shown to improve surgical outcomes, the evidence to date is heterogeneous, without established intervention methods and assessment tools. In gynecologic oncology, only a limited number of prehabilitation studies have been conducted, with only one prospective randomized trial. Further research is warranted to implement an efficient and patient-friendly multimodal prehabilitation service to achieve the best clinical outcomes.

## Figures and Tables

**Figure 1 curroncol-32-00109-f001:**
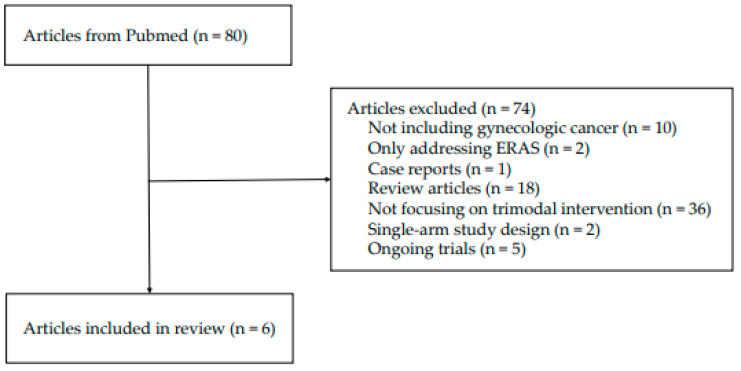
Flow diagram of study selection.

**Table 1 curroncol-32-00109-t001:** Summary of studies on multimodal prehabilitation and surgical outcomes or mortality in gynecologic oncology.

Author (Year) Cancer Type	Study Design Population	Intervention and Duration	Outcome Measures	Results
Diaz-Feijoo (2022) [9] Ovary	Mixed cohort of before-and-after intervention Pilot study N = 34 Prehabilitation: 15, Control: 19	Supervised exercise Nutritional optimization by dietitian (protein supplementation: 1.6–2 g/kg of body weight/day, whey protein, immunomodulatory formula) Psychological preparation ERAS program Duration: 2 (2–3) weeks	Quality of life (QLQ-C30) Physical activity (6MWT, Yale Physical Activity Survey) Nutritional status (Global Leadership Initiative on Malnutrition criteria) HADS	Prehabilitation group: No results available
			Hospital stay	Shorter
			Postoperative complications	NS
			Time to start of chemotherapy	Earlier
Miralpeix (2022) [17] Ovary	Retrospective cohort of consecutive group Pilot study N = 29 NACT/ICS Prehabilitation: 14, Control: 15	Physical exercise recommendations Nutritional counseling (homemade recipes for protein supplementation) Psychological support Preoperative carbohydrate loading Inspiratory threshold-loading device ERAS program Duration: 13.5 ± 2.0 weeks	Mean total protein levels (g/dL): Preoperative Postoperative	Prehabilitation group: Higher Higher
			Intraoperative complications (Clavien–Dindo)	Lower
			Intraoperative blood transfusion	Lower
			First ambulation day, postoperative complications rate, and hospital stay	NS
Lario-Perez (2024) [14] Peritoneum	Prospective cohort N = 62 Ovarian: 30 (48.4%)	Home-based Physical (aerobic and muscular resistance) and relaxation exercises Protein-rich diet and hyperproteic nutritional supplementation (≥1.2 to 1.5 g of protein/kg of body weight/day, immunonutrition supplements) Duration: 27 (19–34) days	Functional walking capacity (T6MWT)	Improved
			Skeletal muscle mass (ASMI)	No improvement
			HADS 90-day postoperative morbidity (Clavien–Dindo II–V)	No results available Only independent risk factor: T6MWT (<360 m)
			Hospital stay	NS between results of T6MWT after prehabilitation
Zębalski (2024) [15] Ovary	Prospective randomized interventional study N = 70 Prehabilitation: 36, Control: 34	Physical activity: resistance and cardio-type aerobic exercises Proper diet and protein supplementation: 1.5–2.0 g/kg of body weight/day, immunomodulators Eliminating risky behaviors and reducing stimulant use: alcohol and smoking cessation Psychological support: refer to psychologist Optimization of laboratory test results ERAS program Duration: 24.5 ± 26.9 days	Physical capacity - 6MWT	Prehabilitation group: Increase
			- VO2 max	No differences
			Maximum muscle tension (LUNA EMG device)	Increase
			Quality of life (standardized EORTC QLQ-C30) HADS Malnutrition (MNA, MUST) Frailty score (G8)	No major impact
			Postoperative complications (Clavien–Dindo)	Fewer
			Hospital stay	Shorter Negative correlation with maximum muscle tension prior to surgery
Miralpeix (2023) [18] Endometrium	Retrospective cohort of consecutive group N = 128 undergoing laparoscopic surgery Prehabilitation: 68, Control: 60	Physical exercise Nutritional counseling Psychological support ERAS program Duration: 25 (18–35) days	Hospital stay	Prehabilitation group: Shorter
			Normal oral diet restart	Earlier
			Postoperative complications (Clavien–Dindo > II)	NS
Li (2024) [16] Gynecology (cervix, endometrium, and ovary)	Prospective cohort of consecutive group N = 97 Prehabilitation: 49, Control: 48	Short-term, hospital-based Exercise intervention (stretching, resistance, aerobic, respiratory function, and Kegel) Nutrition intervention (carbohydrate:protein:fat ratio of 6:3:1, 1.2 g of protein/kg of body weight/day, whey protein, dietitian) Coping strategies to reduce anxiety (yoga relaxation and psychological consultant) ERAS program Duration: 7.2 ± 1.6 days	6MWT Psychological status (HAS) Overall health (RAND 12-Item Health Survey v2	Prehabilitation group: Better Better Better on 1 day before surgery and 30th day after surgery
			Short-term postoperative recovery quality (QoR-9)	Higher quality of recovery for three consecutive days
			Postoperative first ambulation time	Earlier
			First gastrointestinal exhaust time	Earlier
			Hospital stay	NS

Numerical data are given as percentages, median (IQR), or mean ± standard deviation. (6MWT = 6 min walk test; ASMI = appendicular skeletal muscle index; EORTC QLQ-C30 = European Organization for Research and Treatment of Cancer quality of life questionnaire; EMG = electromyography; ERAS = enhanced recovery after surgery; G8 = geriatric 8; HADS = hospital anxiety and depression scale; HAS = hospital anxiety scale; ICS = interval cytoreductive surgery; MNA = mini nutritional assessment; MUST = malnutrition universal screening tool; NACT = neoadjuvant chemotherapy; NS = not significant; QoR-9 = quality of recovery-9; T6MWT = 6 min walk test performed on treadmill; VO2 max = maximal oxygen consumption).

**Table 2 curroncol-32-00109-t002:** Ongoing randomized controlled trials of prehabilitation for surgical treatment of gynecologic cancer.

Study Cancer Type	Objective	N	Intervention and Duration	Outcomes
PROPER—PRehabilitatiOn Plus Enhanced Recovery after surgery versus enhanced recovery after surgery in gynecologic oncology: a randomized clinical trial (NCT04596800) [39] Gynecologic cancer	To evaluate the impact of a prehabilitation program on postoperative recovery time for patients who will undergo gynecologic surgery, following the ERAS guidelines	194 (1:1)	Physiotherapy: aerobic, inspiratory, and stretching exercises, muscle strengthening, 135 min per week in three sessions Nutrition: hypercaloric and hyperproteic nutritional supplement, or whey protein 1 or 2 times/day if indicated; counseling Psychology: counseling, image-based exercises and visualization for relaxation, breathing exercises Duration: 2–3 weeks	Primary: time between surgery and day patient is ready for discharge Secondary: compliance to ERAS guidelines, postoperative complication rates, rates of ICU admissions, health-related QOL, and changes in functional capacity, muscle strength, body mass index, and patient anxiety and depression
Prehabilitation in Gynaecological Cancer Patients (PHOCUS) (NCT04789694) [40] Gynecologic cancer	To evaluate the role of multimodal prehabilitation in patients with gynecological cancer	64 (1:1)	Physical activity: individualized home-based resistance exercises, step count increase by 20% by time of surgery Psychological and nutritional support Duration: 9–12 weeks Recruitment: 36 months	Primary: 6MWT shortly before surgery Secondary: postoperative morbidity, length of postoperative hospital stay, adherence to training program, effects of nutritional support (analyzed from blood), QOL, and psychological health
TRAINING-Ovary 01 (connecTed pRehabiliAtIoN pelvIc caNcer surGery): a multicenter randomized study comparing neoadjuvant chemotherapy for patients managed for ovarian cancer with or without a connected prehabilitation program (NCT04451369) [41] Ovary	To determine if a connected prehabilitation program during NACT for patients treated for an advanced ovarian cancer will improve physical capacity before major abdomino-pelvic surgery	136 (1:1)	Physical activity training program at home: short videos available on smartphone app, daily physical activity sessions and weekly supervision with connected device Nutritional care (ESPEN guidelines) adapted according to information transmitted weekly Psychological support with coping strategies Recruitment: 24 months Follow up: 5 years	Primary: variation in VO2 max between baseline and surgery Secondary: compliance, physical and nutritional status, QOL, morbidity, postoperative outcomes, return to intended oncologic treatments, oncologic outcomes, and cost-effectiveness
A multimodal prehabilitation program for the reduction of postoperative complications after surgery in advanced ovarian cancer under an ERAS pathway: a randomized multicenter trial (SOPHIE) (NCT04862325) [42] Ovary	To compare the postoperative complications of a multimodal prehabilitation program in patients undergoing cytoreductive surgery for advanced ovarian cancer with standard preoperative care	146 (1:1)	Physical interventions: supervised high-intensity training program (endurance and resistance training), inspiratory muscle training program using threshold resistance device, promotion of physical activity through health mobile application Psychological and cognitive behavioral therapy: weekly group-based sessions Nutritional intervention: protein goals (1.6–2 g/kg/day), whey protein isolate, immunomodulatory formula Exclusion: patients with <75% adherence Duration: >2 weeks Recruitment: 4 years	Primary: overall postoperative complication rate (CCI) Secondary: length of hospital stay and days until initiation of chemotherapy; baseline, preoperative, and 1-month postoperative QOL; physical, nutritional, and cognitive test assessments; and prehabilitation and ERAS program compliance

6MWT = 6 min walk test; CCI = comprehensive complication index; ERAS = enhanced recovery after surgery; ESPEN = European Society for Clinical Nutrition and Metabolism; ICU = intensive care unit; NACT = neoadjuvant chemotherapy; QOL = quality of life; VO2 max = maximum oxygen uptake.

## Data Availability

No new data were created or analyzed in this study. Data sharing is not applicable to this article.

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
