# Peer review of "Multimodal Prehabilitation for Gynecologic Cancer Surgery"

_curroncol, 2025, doi:10.3390/curroncol32020109_

Round 1

Reviewer 1 Report

Comments and Suggestions for Authors

Dear authors thank you for your effort that address this current issue in the management of patients with gynecologic cancer.

The main problems of prehabilitation programs are compliance and acceptance of patients.  Also there are no any standardized methods that evaluate effectiveness of this program and many variables such as compliance, acceptance, intervention period, type of surgical procedure and result evaluation methods may affect the results. All of these variables may cause bias. 

At line  147-148  …… decreased by 51.49 m   ….. may be increased by 51.49

At line 250-251, the sentence should be checked.

Author Response

Thank you for your insightful and thoughtful comments. We have carefully addressed your feedback and made the necessary corrections as suggested.

Reviewer #1:

  1. At line 147-148 …… decreased by 51.49 m ….. may be increased by 51.49

Response: Thank you for your review. According to the referenced study, the term "decreased" is indeed correct, potentially due to surgical stress. The authors aimed to demonstrate that the intervention group increased their walking capacity before surgery, thereby somewhat compensating for the anticipated postoperative decline (preop +38.27 m → postop 30th day -51.49 m, resulting in a total change of -13.23 m). In contrast, the control group showed a total decrease of 81.64 m on the 30th day after surgery compared to baseline (preop -15.15 m → postop 30th day -66.5 m, leading to a total change of -81.64 m).

We have made a slight modification to the immediately preceding sentence for clarification. The original sentence, "The compliance rate was 100%, and superior physical status was reported not only on the day before surgery but also on the 30th day after surgery," has been revised to: "The compliance rate was 100%, and a difference in physical status was reported not only on the day before surgery but also on the 30th day after surgery." (Line 154)

  1. At line 250-251, the sentence should be checked.

Response: We appreciate your valuable suggestion and made the following changes:

The original sentence, "Additionally, two institutions identified in this review conducted group sessions (cognitive behavioral group sessions [9] and supervised group mindfulness hospital sessions [10])," has been revised to:

" Additionally, there were two studies that conducted group sessions for psychological intervention: cognitive behavioral group sessions [9] and supervised group mindfulness hospital sessions [10]." (Lines 265-267)

Reviewer 2 Report

Comments and Suggestions for Authors

Thank you for submitting this comprehensive review of prehabilitation in gynaecological oncology.  This review provides a good summary of previous studies and ongoing trials.

I would ask you to address why your review excludes patients with vulva and vaginal cancer.

In Lines 96-100 unclear what the primary outcome is

Line 116 what does 'inspiration muscle strength' mean?

Line 120 what do you mean by 'articulation'?

Lines 136, 201-202 and 362 please do not say 'peritoneal cancer/carcinomatosis/endometrial cancer patients.  instead say 'patients with ...cancer'

Line 368 what does 'financial toxicity' mean? for the patient or the hospital?

Lines 371-372 need grammatical correction. Unclear point.

Author Response

  1. I would ask you to address why your review excludes patients with vulva and vaginal cancer.

Response: We agree with your comment that prehabilitation should be implemented in other types of gynecologic cancers as well. However, we could not find any publications that specifically address the effects of prehabilitation on surgical treatment for vulvar or vaginal cancer. This may be attributed to the relatively small number of patients with these cancers. Instead, we identified articles that focus on these diseases from different perspectives, which we addressed at lines 406-412.

  1. In Lines 96-100 unclear what the primary outcome is

Response: Thank you for your comment. We have revised the sentence to include the following terms: “postoperative complication rates and length of hospitalization after ovarian cytoreductive surgery” (Lines 106-107).

  1. Line 116 what does 'inspiration muscle strength' mean?

Response: We apologize for the spelling error. The word ‘inspiration’ should be ‘inspiratory’. The word has been revised accordingly (Line 125).

  1. Line 120 what do you mean by 'articulation'?

Response: We tried to elaborate on all intervention strategies introduced in the cited articles to the full extent. One Chinese study mentioned, "warm-up before exercise was performed for 5 min, including ankle raising, stretching, and articulation". Since there was no additional description of this method in the original reference, we decided to remove this word in the manuscript (Line 128).

  1. Lines 136, 201-202 and 362 please do not say 'peritoneal cancer/carcinomatosis/endometrial cancer patients. instead say 'patients with ...cancer'

Response: Thank you for your suggestion. As you’ve suggested, we replaced the terms 'peritoneal cancer/carcinomatosis/endometrial cancer patients' with 'patients with ... cancer.'

  1. Line 368 what does 'financial toxicity' mean? for the patient or the hospital?

Response: Financial toxicity refers to that of both patients’ and the society’s. We revised the sentence to clarify the meaning (Lines 384-385).

  1. Lines 371-372 need grammatical correction. Unclear point.

Response: We concur with your feedback and appreciate your input. The relevant sentence has been revised as follows: "They showed a favorable result of prehabilitation by calculating the cost-effectiveness of up to $9,418 per patient" (Lines 388-389).

Reviewer 3 Report

Comments and Suggestions for Authors

This is a well written narrative review summarizing the current evidence on rehabilitation programs in gynecologic oncology. I have several comments for the authors:

Introduction:

-       Line 28 – “Among them, gynecologic…portion”. I suggest to be specific and highlight the % of gynecologic cancer out of all cancers.

Methods:

-       The authors should specify the years that were included in PubMed search. They commented that they included “recent studies”, but need to be specific in terms of range of years included.

-       Was PRISMA guidelines followed for this review? If yes, need to add that to the methods part. If not, need to clarify why not.

Results:

-       I recommend adding a figure with flow diagram that describes total number of initial studies reviewed and reason for exclusion.

-       Is information on time from prehab to surgery available? If yes, please add that to table 1

-       Although this was not part of the studies included in the review, I recommend adding a paragraph under “nutritional intervention” section, on the evidence of giving TPN in this setting and discussing if there are any benefits

-       Any data on discharge destination? Were these patients discharged home vs rehab? Did prehab programs influenced the likelihood of these patients to be discharged home? Please add a paragraph on that in the results and it is an important outcome as well

Author Response

Introduction:

-       Line 28 – “Among them, gynecologic…portion”. I suggest to be specific and highlight the % of gynecologic cancer out of all cancers.

Response: Thank you for your helpful comment. We added the specific data within the sentence as suggested (Line 29).

Methods:

-       The authors should specify the years that were included in PubMed search. They commented that they included “recent studies”, but need to be specific in terms of range of years included.

Response: We appreciate your insightful comment, as it has allowed me to incorporate more precise wording. We have included the exact duration ("from the database's inception through November 2024") and have revised the sentence for greater clarity (Lines 73-74).

-       Was PRISMA guidelines followed for this review? If yes, need to add that to the methods part. If not, need to clarify why not.

Response: It would’ve been helpful if this review had been designed as a systematic review according to the PRISMA guidelines. However, we found that the number of studies targeting prehabilitation for gynecologic cancer surgery is very small and with significant heterogeneity. Therefore, this review was designed as a narrative review to include and discuss relevant studies as much as possible.

Results:

-       I recommend adding a figure with flow diagram that describes total number of initial studies reviewed and reason for exclusion.

Response: Thank you for the comment. A figure with a flow diagram was added (Figure 1).

-       Is information on time from prehab to surgery available? If yes, please add that to table 1

Response: Thank you for your thoughtful question. The articles that provided the duration of intervention are shown in Table 1. If you are referring to the period from the completion of prehabilitation to surgery, no study has specifically addressed this timeframe. Generally, prehabilitation aims to enhance the functional capacity prior to surgery, and there was no mention of surgery being postponed to achieve optimal functional capacity in the reviewed articles.

-       Although this was not part of the studies included in the review, I recommend adding a paragraph under “nutritional intervention” section, on the evidence of giving TPN in this setting and discussing if there are any benefits

Response: Thank you for your comment. This information has been added to lines 216-222, as follows:

“Interestingly, all nutritional interventions to date have included only supplementation or diet coaching. However, for gynecologic cancer patients for whom enteral nutrition is un-suitable at the time of initiating cancer treatment, total parenteral nutrition (TPN) may be beneficial as a means of prehabilitation as well [21]. This is particularly the case in advanced ovarian cancer patients since they have a high prevalence of malnutrition. However, the current evidence regarding the use of TPN as a process of prehabilitation is lacking.”

-       Any data on discharge destination? Were these patients discharged home vs rehab? Did prehab programs influenced the likelihood of these patients to be discharged home? Please add a paragraph on that in the results and it is an important outcome as well

Response: As mentioned from line 386, Dholakia et al. [36] addressed this issue and demonstrated the cost-effectiveness of reducing non-home discharge rates using a hypothetical cohort. However, the actual benefits regarding non-home discharge rates are yet to be discovered in the studies included of this review, as well as in the ongoing trials. Your points are highly valuable and need to be considered in future research, but the results to date lack sufficient substance for definitive conclusions. This point has been mentioned in lines 393-397.